# Adaptive MCMC for Bayesian Variable Selection in Generalised Linear Models and Survival Models

**DOI:** 10.3390/e25091310

**Published:** 2023-09-08

**Authors:** Xitong Liang, Samuel Livingstone, Jim Griffin

**Affiliations:** Department of Statistical Science, University College London, London WC1E 6BT, UK; samuel.livingstone@ucl.ac.uk (S.L.); j.griffin@ucl.ac.uk (J.G.)

**Keywords:** Bayesian computation, Bayesian variable selection, spike-and-slab priors, adaptive Markov Chain Monte Carlo, generalised linear models, survival models

## Abstract

Developing an efficient computational scheme for high-dimensional Bayesian variable selection in generalised linear models and survival models has always been a challenging problem due to the absence of closed-form solutions to the marginal likelihood. The Reversible Jump Markov Chain Monte Carlo (RJMCMC) approach can be employed to jointly sample models and coefficients, but the effective design of the trans-dimensional jumps of RJMCMC can be challenging, making it hard to implement. Alternatively, the marginal likelihood can be derived conditional on latent variables using a data-augmentation scheme (e.g., Pólya-gamma data augmentation for logistic regression) or using other estimation methods. However, suitable data-augmentation schemes are not available for every generalised linear model and survival model, and estimating the marginal likelihood using a Laplace approximation or a correlated pseudo-marginal method can be computationally expensive. In this paper, three main contributions are presented. Firstly, we present an extended *Point-wise implementation of Adaptive Random Neighbourhood Informed* proposal (PARNI) to efficiently sample models directly from the marginal posterior distributions of generalised linear models and survival models. Secondly, in light of the recently proposed approximate Laplace approximation, we describe an efficient and accurate estimation method for marginal likelihood that involves adaptive parameters. Additionally, we describe a new method to adapt the algorithmic tuning parameters of the PARNI proposal by replacing Rao-Blackwellised estimates with the combination of a warm-start estimate and the ergodic average. We present numerous numerical results from simulated data and eight high-dimensional genetic mapping data-sets to showcase the efficiency of the novel PARNI proposal compared with the baseline add–delete–swap proposal.

## 1. Introduction

Variable selection is an automatic method for finding a small subset of covariates that explain most of the variation in the response of interest. In addition to identifying the most predictive covariates, there is a growing interest in exploring the low-rank structure between the covariates and the response, especially in genetic mapping problems where the objective is to find the expressed genes that are associated with a specific disease the most. In the frequentist framework, model selection is based on maximising the penalised log-likelihood [1] or minimising information criteria such as AIC [2] and BIC [3]. Other approaches, such as the deviance information criterion (DIC) [4] and widely applicable information criterion (WAIC) [5], which are generalisations of the AIC, are also popular in model selection.

A natural alternative to these frequentist approaches is Bayesian variable selection (BVS). In the Bayesian approach, a prior is imposed on all candidate models, and the resulting posterior distribution naturally captures model uncertainty. In this work, we consider a spike-and-slab prior [6,7], which introduces indicator variables denoting the inclusion or exclusion of every covariate. Therefore, the spike-and-slab prior leads to a model posterior distribution that lies in a lattice with the same dimension as the number of covariates. We can understand the dependency between the importance of covariates and response using natural measures of the posterior distribution such as posterior model probability (PMP) and marginal posterior inclusion probability (PIP). The computation of the exact posterior distribution requires a full search over the whole model space, which is computationally infeasible when a high-dimensional data-set is analysed. In these settings, Markov Chain Monte Carlo (MCMC) algorithms are often used to explore the model space and estimate the posterior distribution. For “large *n*, large *p*” data-sets, which are now often encountered in some problems in genetics/genomics (such as genetic mapping studies), such algorithms must be carefully designed. In this work, we mainly consider Bayesian variable selection in generalised linear models and survival models and focus on three popular models: the logistic regression model [8,9], the Cox proportional hazards model with partial likelihood [10,11,12,13,14] and the Weibull regression model [15]. In each case, we illustrate how carefully designed algorithms can facilitate effective posterior computation.

A natural challenge of Bayesian variable selection methods in the above settings is that the marginal likelihood (or the integrated likelihood in [16]) is not analytically available. One set of solutions are Reversible Jump MCMC schemes (RJMCMC) [17], which sample from the joint space of models and regression coefficients by jointly proposing moves between models and regression coefficients. But it is often difficult to construct efficient proposals for these trans-dimensional jumps and design an MCMC scheme that mixes well [18]. For some specific models, data-augmentation methods [19] are available and result in closed-form marginal likelihood conditioned on latent variables, for instance, Pólya-gamma data augmentation [20] for logistic regression. For other models where no suitable data-augmentation scheme exists, the most popular approaches are the Laplace approximation and the correlated pseudo-marginal method [21], which rely on finding the maximum a posteriori (MAP) estimate of the regression coefficients. A novel scalable estimation method for marginal likelihood, approximate Laplace approximation (ALA), is introduced in [16] and relies on defining an initial value for the coefficient parameters. ALA can save computational time during the optimisation process of finding the MAP estimate, but it does not yield an asymptotically consistent estimate. A detailed discussion of these approaches will be given in Section 3.

Assuming that the marginal likelihood has been estimated, several MCMC algorithms can be used for simulation starting from the posterior distribution of BVS. The widely used add–delete–swap proposal [22] can be employed here. The add–delete–swap proposal generates a new model by randomly selecting one of three possible moves: addition/deletion of a covariate into/from the model or swapping one covariate that is included with another that is not. Although it has been proved in [23] that the add–delete–swap proposal can produce a rapidly mixing Markov Chain, the chains may still converge slowly, particularly when dealing with large-*p* problems. Adaptive MCMC schemes [24], which involve updating tuning parameters on the fly, are found to be valuable in addressing the issue of poor convergence. Lamnisos et al. [25] describe an adaptive add–delete–swap proposal that allows for simultaneous changes to multiple variables at a time. Griffin et al. [26] introduce the Adaptively Scaled Individual adaptation proposal (ASI), which simulates a new model with probability proportional to the product of PIPs. Wan and Griffin [27] extend the ASI proposal to logistic regression and accelerated failure time models. Other popular MCMC approaches include the Hamming ball sampler (HBS) [28], which proposes a new model within a Hamming neighbourhood using the PMPs as proposal weights, and the tempered Gibbs sampler [29,30], which uses tempering to efficiently sample from the multi-modal posteriors that commonly arise due to highly correlated covariates.

The recent work of [31] provides useful insights into the design of efficient MCMC schemes in discrete spaces. The work introduces the locally informed proposal, which re-weights a given non-informed base kernel with a function of the PMPs. It is shown in [31] that the locally informed proposal constructed with a balancing function that satisfies certain functional properties is asymptotically optimal compared with other choices of function in terms of Peskun ordering. Building upon the idea of locally informed proposal, Zhou et al. [32] show that the locally informed and thresholded (LIT) proposal can achieve dimension-free mixing times under conditions similar to those mentioned in [23] for BVS in linear regression models. Recent work in [33] introduces a Point-wise implementation of the Adaptive Random Neighbourhood Informed proposal (PARNI), which combines the advantages of both adaptive schemes and locally informed proposals. The PARNI proposal outperforms other state-of-the-art algorithms in a wide range of high-dimensional data-sets for BVS in linear regression models.

Other computational approaches are also available for estimating the BVS posterior distribution. Hans et al. [34] introduce a novel Shotgun Stochastic Search (SSS) approach that also explores the “local neighbourhood” idea and targets very high-dimensional model spaces to find high-probability regions. The integrated nested Laplace approximations (INLAs) [35] can solve latent Gaussian models including generalised linear models and approximate the posterior marginals obtained from the continuous priors [36]. Sara et al. [37] view survival models as latent Gaussian models and also approximate the posterior marginals using INLAs. The posterior distribution can also be approximated using Variational Bayes (VB) [38]. Ray et al. [39] describe a scalable mean-field variational family to approximate the posterior distribution of BVS in linear regression and extended this VB approximation to the logistic regression model in [40]. Komodromos et al. [41] apply the Sparse Variational Bayes (SVB) method to approximate the posterior of proportional hazards models with partial likelihood. Other works develop a sampling strategy based on simulating piece-wise deterministic Markov processes (PDMPs) [42,43], which directly target the posterior distribution obtained from a spike-and-slab prior.

In this paper, we extend the PARNI proposal to sampling from the BVS posterior distribution in generalised linear models and survival models. To avoid the overwhelming computational costs of approximating the marginal likelihood in the locally informed proposals and motivated by ALA [16], we introduce an ALA estimate of the marginal likelihood with a novel initial value. In contrast to the suggestion in [16], which initialises ALA at origin, the novel initial value is adaptively updated on the fly using previously sampled models. The new method is computationally less complex than the Laplace approximation or correlated pseudo-marginal scheme as a result of avoiding iterative optimisation and provides a more accurate estimate than the original ALA approach initialised at the origin. We also consider new approaches to adapt the tuning parameters in the PARNI proposal. The new adaptation scheme replaces the Rao-Blackwellised estimates of PIPs using the combination of a warm-start estimate and the ergodic average calculated using previously sampled models.

To illustrate the performance of the new PARNI scheme in real-life high-dimensional problems, we perform BVS on eight genetic mapping data-sets (four for the logistic regression model and four for survival analysis) and compare the output of the PARNI proposal with the add–delete–swap proposal as a baseline. For the logistic model with binary outcome, we consider the problem of finding expressed genes that are related the most to the presence of Systemic Lupus Erythematosus in a case-control study with 10,995 observations and various numbers of SNPs, from 5771 to 42,430, on four different chromosomes. In survival analysis, we consider four cancer-related data-sets (two for breast cancer and two for lung cancer), containing patients ranging from 130 to 1904 and genetic covariates varying from 662 to 54,675.

This paper is organised as follows: In Section 2, we review the model setup and prior specification for BVS in generalised linear models, Cox proportional hazards and Weibull survival models. In Section 3, we introduce four computational methods to estimate the marginal likelihood. Section 4 describes the PARNI proposal, highlighting the novelties in the adaption of algorithmic tuning parameters and the calculation of the accurate and efficient marginal likelihood estimates. We implement these MCMC algorithms in Section 5 and compare their performance with the add–delete–swap proposal on several real data-sets. We include a discussion in Section 6, highlighting some possible future research directions.

## 2. Bayesian Variable Selection for Generalised Linear Models and Survival Models

### 2.1. Generic Model Setting

Suppose that *p* covariates are available in the data. Let $X={\left(x_{1},\cdots ,x_{n}\right)}^{T}\in R^{n\times p}$ be the full data matrix that contains *n* observations with rows $x_{i}=\left(X_{i1},\cdots ,X_{ip}\right)$ and let $Z={\left(z_{1},\cdots ,z_{n}\right)}^{T}\in R^{n\times q}$ be the full data matrix that contains *q* variables that must be included in every model. Let binary vector $\gamma =\left(\gamma_{1},\cdots ,\gamma_{p}\right)\in \Gamma ={\left\{0,1\right\}}^{p}$ be a model indicator, where $\gamma_{j}=1$ if the *j*-th variable is included in model $M_{\gamma }$ and $\gamma_{j}=0$ otherwise.

Let $y=(y_{1},\cdots ,y_{n})$ be the vector of responses. The generalised linear model associated with model $M_{\gamma }$ can be specified as
(1)$$\begin{array}{c} y_{i}∼F\left(\mu_{\gamma ,i},\phi \right) \end{array}$$
where F(μ,ϕ) is a distribution that belongs to the exponential family with mean μ and dispersion ϕ. Linear predictor $\eta_{\gamma ,i}$ is defined as
(2)$$\begin{array}{c} \eta_{\gamma ,i}=z_{i}^{T}\alpha +x_{\gamma ,i}^{T}\beta_{\gamma } \end{array}$$
where $x_{\gamma ,i}$ contains those variables *j* for which $\gamma_{j}=1$. In addition, we define the size of model $M_{\gamma }$ as $p_{\gamma }=\sum_{j=1}^{p}\gamma_{j}$. Linear predictor $\eta_{\gamma ,i}$ is mapped to mean $\mu_{\gamma ,i}$ using link function *g* as
(3)$$\begin{array}{c} \eta_{\gamma ,i}=g\left(\mu_{\gamma ,i}\right). \end{array}$$

We consider the following setup of survival models: For the *i*-th patient, given hazard function $h_{i}\left(t\right)$ at time *t*, the probability that an event occurs at time $T_{i}$ before a certain time $t_{i}$ can be written as
$$\begin{array}{cc} F_{T_{i}}\left(t\right) & =P\left(T_{i}\leq t_{i}\right)=1-S_{T_{i}}\left(t_{i}\right) \end{array}$$
where $S_{T_{i}}$ is called the survival function and is defined by
$$\begin{array}{c} S_{T_{i}}\left(t\right)=\exp\left\{-\int_{0}^{t}h_{i}\left(u\right)du\right\}. \end{array}$$
Data often involve censoring where the true time to event is not observed. Let *t* be the vector of the observed times, where each $t_{i}$ denotes the minimum of censoring time $C_{i}$ and survival time $T_{i}$. In the case of “right-censored” data, we define an *n*-dimensional event indicator vector *d* to denote, for each patient *i*, whether the event was observed during their follow-up ($d_{i}=1$) or was censored ($d_{i}=0$). In the case where the event was observed for patient *i* ($d_{i}=1$), then $t_{i}$ denotes their time to event; otherwise, we observed the length of their follow-up.

Given a model $M_{\gamma }$ associated with linear predictor $\eta_{\gamma ,i}$ as in (Equation 2), we consider the exponential hazard, $\lambda_{\gamma ,i}=\exp\left(\eta_{\gamma ,i}\right)$, and assume that the hazard function conditioned on model $M_{\gamma }$ has the form
(4)$$\begin{array}{c} h_{\gamma ,i}\left(t\right)=h\left(t,\lambda_{\gamma ,i},k\right) \end{array}$$
where *k* is an additional shape parameter if needed. We can conclude the following log-likelihood on y=(t,d):$$\begin{array}{c} \log p\left(y|\alpha ,\beta_{\gamma },\gamma \right)=\sum_{i=1}^{n}d_{i}\log\left(h_{\gamma ,i}\left(t_{i}\right)\right)-H_{i}\left(t_{i}\right) \end{array}$$
where $H_{i}$ is the cumulative hazard function for the *i*-th patient and is defined by
$$\begin{array}{c} H_{i}\left(t\right)=\int_{0}^{t}-\frac{d\log S(t)}{dt}{|}_{t=u}du=-\log S_{T_{i}}\left(t\right). \end{array}$$

### 2.2. Prior Elicitation

Recalling model indicator $\gamma \in {\left\{0,1\right\}}^{p}$, we consider the prior structure
(5)$$\begin{array}{c} p\left(\alpha ,\beta_{\gamma },\phi ,\gamma \right)\propto p\left(\alpha \right)p\left(\beta_{\gamma }|\phi ,\gamma \right)p\left(\phi |\gamma \right)p\left(\gamma \right). \end{array}$$

For generalised linear models in which the dispersion parameter is known (e.g., the logistic regression model where ϕ=1) or some survival models that do not involve a dispersion parameter, the prior specification becomes
(6)$$\begin{array}{c} p\left(\alpha ,\beta_{\gamma },\gamma \right)\propto p\left(\alpha \right)p\left(\beta_{\gamma }|\gamma \right)p\left(\gamma \right), \end{array}$$
which is equivalent to treating ϕ as a fixed parameter. In this work, we focus on the prior structure described in (Equation 6), and we assume that there is no additional dispersion parameter in the model.

We specify the following prior distribution for the coefficient parameters:(7)$$\begin{array}{c} \begin{array}{cc} \alpha & ∼N_{q}\left(0,\sigma_{\alpha }^{2}I_{q}\right)\\ \beta_{\gamma }|\gamma & ∼N_{p_{\gamma }}\left(0,gI_{p_{\gamma }}\right) \end{array} \end{array}$$
where *g* is a positive scale parameter, $\sigma_{\alpha }^{2}$ is the prior variance on the coefficients of the fixed covariates, $I_{p}$ denotes a p×p identity matrix and $p_{\gamma }=\sum_{j=1}^{p}\gamma_{j}$ is the size of model $M_{\gamma }$.

We consider the choice of model prior
(8)$$\begin{array}{c} p\left(\gamma \right)=h^{p_{\gamma }}{\left(1-h\right)}^{p-p_{\gamma }} \end{array}$$
where hyper-parameter *h* denotes the prior inclusion probability for each variable.

It is possible to construct a fully Bayesian hierarchical model based on the prior specifications described above. We can impose the following hyper-priors on hyper-parameters *g* and *h*:$$\begin{array}{cc} \sqrt{g} & ∼C^{+}\left(0,1\right)\\ h & ∼\mathrm{Beta}\left(a,b\right) \end{array}$$
where $C^{+}\left(0,1\right)$ denotes the standard half-Cauchy distribution and Beta(a,b) denotes the Beta distribution with parameters a>0 and b>0. The half-Cauchy hyper-prior is a generalisation of the horseshoe prior [44,45,46], employed on the global-scale parameter of a continuous mixture of normal priors, to BVS problems. Liang et al. [47] note that fixing *g* can lead to several paradoxes and problems of model mis-specification. For other possible choices of hyper-priors on *g*, see [47,48]. In the context of prior inclusion probability *h*, Ley et al. [49] advise against using a fixed *h* in the absence of strong prior knowledge about the number of important variables. Kohn et al. [50] propose a Beta-binomial model prior in which hyper-parameter *h* can be integrated out analytically, leading to
$$\begin{array}{c} p\left(\gamma \right)=\frac{B\left(a+p_{\gamma },b+p-p_{\gamma }\right)}{B\left(a,b\right)} \end{array}$$
where B(·,·) denotes the Beta function.

### 2.3. Logistic Regression

Assume that $y_{i}\in \left\{0,1\right\}$, with $y_{i}=1$ indicating the success of an event and $y_{i}=0$ indicating failure. Logistic regression links the proportion of successes to the linear predictor with a logistic link function *g* as
(9)$$\begin{array}{c} \eta_{\gamma ,i}=\log\left(\frac{\mu_{\gamma ,i}}{1-\mu_{\gamma ,i}}\right)=z_{i}^{T}\alpha +x_{\gamma ,i}^{T}\beta_{\gamma },\;i=1,\cdots ,n, \end{array}$$
and the response variable is modelled as $y_{i}∼\mathrm{Bern}\left(\mu_{\gamma ,i}\right)$ under model $M_{\gamma }$.

### 2.4. Cox Proportional Hazards (PHs) with Partial Likelihood

Starting with exponential hazard function $\lambda_{\gamma ,i}=\exp\left(\eta_{\gamma ,i}\right)$ associated with model $M_{\gamma }$, in the Cox proportional hazard function, the hazards are assumed to have the form
(10)$$\begin{array}{c} h_{i}\left(t\right)=h_{0}\left(t\right)\lambda_{\gamma ,i} \end{array}$$
where $h_{0}$ is some baseline hazard function. In this proportional model, all covariate effects are assumed to be multiplicative. The full likelihood is then given as
(11)$$\begin{array}{c} L\left(\alpha ,\beta_{\gamma },H_{0}|y,\gamma \right)\propto \prod_{j=1}^{n}{\left(\exp\left(\eta_{\gamma ,i}\right)H_{0}^{\prime }\left(t_{i}\right)\right)}^{d_{i}}\exp\left\{-\exp\left(\eta_{\gamma ,i}\right)H_{0}\left(t_{i}\right)\right\} \end{array}$$
where $H_{0}\left(t\right)=\int_{0}^{t}h_{0}\left(u\right)du$ is the cumulative baseline hazard function. If we model the prior of $H_{0}$ with a prior process $p(H_{0})$ on the cumulative hazard function, the resulting posterior distribution of α and $\beta_{\gamma }$ is
(12)$$\begin{array}{c} p\left(\alpha ,\beta_{\gamma }|y,\gamma \right)\propto \int L\left(\alpha ,\beta_{\gamma },H_{0}|y,\gamma \right)\times p\left(\alpha \right)p\left(\beta_{\gamma }|\gamma \right)p\left(H_{0}\right)dH_{0}. \end{array}$$
Alternatively, we can take the partial likelihood of Cox, which is given by
(13)$$\begin{array}{c} \mathrm{PL}\left(\alpha ,\beta_{\gamma }|y,\gamma \right)\propto \prod_{i=1}^{n}{\left\{\frac{\exp(\eta_{\gamma ,i})}{\sum_{s\in R(t_{i})}\exp\left(\eta_{\gamma ,s}\right)}\right\}}^{d_{i}} \end{array}$$
where $R\left(t\right)=\left\{i:\;t_{i}\geq t\right\}$ is the set of patients at risk at time *t*. Unlike the full likelihood formulated in (Equation 11), partial likelihood does not rely on the specification and estimation of baseline hazard function $h_{0}$. Partial likelihood and its variants are, therefore, popular alternatives to full likelihood in many survival studies [11,14,51]. It is highlighted in [52,53] that partial likelihood can be obtained by integrating out the baseline hazard function using a Gamma process prior. Bayesian inference with partial likelihood (Equation 13) relies on approximate posterior $p_{\mathrm{PL}}\left(\alpha ,\beta_{\gamma }|y,\gamma \right)$, which can be expressed as
(14)$$\begin{array}{c} p_{\mathrm{PL}}\left(\alpha ,\beta_{\gamma }|y,\gamma \right)\propto \mathrm{PL}\left(\alpha ,\beta_{\gamma }|y,\gamma \right)\times p\left(\alpha \right)p\left(\beta_{\gamma }|\gamma \right) \end{array}$$
where baseline hazard function $h_{0}$ is eliminated.

### 2.5. Weibull Regression

In addition to the semi-parametric approach of Cox PHs with partial likelihood, we consider another commonly used parametric model for survival analysis, namely, the Weibull model. A Weibull model is obtained by extending the exponential model by raising the survival rate to a positive power *k*, giving
(15)$$\begin{array}{c} S_{i}\left(t\right)=\exp\left\{-{\left(t\lambda_{i}\right)}^{k}\right\}. \end{array}$$
Parameter *k* is the shape parameter of a Weibull random variable. When k<1, the hazard rate decreases over time. Conversely, when k>1, the hazard rate increases over time. It is possible to recover the exponential survival model when k=1, and it represents a constant hazard rate over time.

We can derive the hazard function
(16)$$\begin{array}{c} h_{i}\left(t\right)=-\frac{d}{dt}\log\left(S_{i}\left(t\right)\right)=\lambda_{i}k{\left(\lambda_{i}t\right)}^{k-1} \end{array}$$
and the log-likelihood for parameters α, $\beta_{\gamma }$ and *k* as
(17)$$\begin{array}{c} \log\left(L\left(\alpha ,\beta_{\gamma },k|y,\gamma \right)\right)=\sum_{i=1}^{n}d_{i}\left[\log\left(k\right)+k\log\left(\lambda_{i}\right)+\left(k-1\right)\log\left(t_{i}\right)\right]-{\left(t_{i}\lambda_{i}\right)}^{k}. \end{array}$$

It should be noted that the Weibull distribution does not belong to the exponential family, unless shape parameter *k* is assumed to be fixed. In the Bayesian framework, we consider the prior $p\left(\log\left(k\right)\right)=N\left(0,\sigma_{k}^{2}\right)$ for some positive prior variance $\sigma_{k}^{2}$ as in [15]. To perform MCMC, we alternatively update γ|k using the PARNI proposal and k|γ using an adaptive random walk proposal as described in Appendix B.

## 3. Computation of Marginal Likelihood p(y|γ)

Let $\theta_{\gamma }=\left(\alpha ,\beta_{\gamma }\right)$ be the collection of all coefficient parameters associated with model $M_{\gamma }$. We are interested in simulating samples from the posterior distribution π(γ)∝p(y|γ)p(γ), where p(y|γ) represents the marginal likelihood, given by
(18)$$\begin{array}{c} p\left(y|\gamma \right)\propto \int p\left(y|\theta_{\gamma },\gamma \right)p\left(\theta_{\gamma }|\gamma \right)d\theta_{\gamma }. \end{array}$$
In generalised linear models and survival analysis, a closed-form solution to (Equation 18) is typically not analytically available.

Assuming that an estimate of marginal likelihood $\hat{p}\left(y|\gamma \right)$ can be obtained, we consider MCMC algorithms with random neighbourhood proposals as described in [33], which is a sub-class of Metropolis–Hastings (MH) schemes [54,55]. The random neighbourhood proposal consists of the following three stages:Around the current model, γ, randomly generate a neighbourhood N∼p(·|γ).Propose a new model, $\gamma^{\prime }$, within random neighbourhood N according to $q_{N}\left(\gamma ,\cdot \right)$.Accept the new proposal, $\gamma^{\prime }$, with the MH acceptance probability
(19)$$\begin{array}{cc} \alpha (\gamma ,\gamma^{\prime }) & =\min\left\{1,\frac{\pi \left(\gamma^{\prime }\right)p\left(N^{\prime }|\gamma^{\prime }\right)q_{N^{\prime }}\left(\gamma^{\prime },\gamma \right)}{\pi \left(\gamma \right)p\left(N|\gamma \right)q_{N}\left(\gamma ,\gamma^{\prime }\right)}\right\}\\ \; & =\min\left\{1,\frac{\hat{p}\left(y|\gamma^{\prime }\right)p\left(\gamma^{\prime }\right)p\left(N^{\prime }|\gamma^{\prime }\right)q_{N^{\prime }}\left(\gamma^{\prime },\gamma \right)}{\hat{p}\left(y|\gamma \right)p\left(\gamma \right)p\left(N|\gamma \right)q_{N}\left(\gamma ,\gamma^{\prime }\right)}\right\} \end{array}$$where $N^{\prime }$ is the neighbourhood used in the reverse move of the MH scheme.

In this section, we will describe four methods commonly used to estimate marginal likelihood p(y|γ): data augmentation, Laplace approximation, correlated pseudo-marginal and approximate Laplace approximation. Before introducing these methods, it is necessary to define the following terms for convenience. Let $J_{\gamma }$ be a $n\times (q+p_{\gamma })$ matrix which contains all necessary covariates for model $M_{\gamma }$ and is given by $J_{\gamma }=\left(Z\;X_{\gamma }\right)$, and let $V_{\gamma }$ be the variance–covariance matrix of the prior distribution of $\theta_{\gamma }$, defined by
(20)$$\begin{array}{c} V_{\gamma }=\left(\begin{array}{cc} \sigma_{\alpha }^{2}I_{q} & 0\\ 0 & gI_{p_{\gamma }} \end{array}\right). \end{array}$$

### 3.1. Data Augmentation

The data-augmentation scheme [19] introduces latent variables ω into the model such that the posterior distribution of variables of interest becomes analytically tractable given ω. The Pólya-gamma data-augmentation scheme [20] can be utilised for the logistic regression model to evaluate the marginal likelihood. Given real numbers ψ∈R, a>0, b>0 and a set of latent variables $\omega =(\omega_{1},\cdots ,\omega_{n})$, in which each individual $\omega_{i}$ follows a Pólya-gamma distribution PG(b,0), the application of Pólya-gamma data augmentation exploits the following identity:(21)$$\begin{array}{c} \frac{{\left(\exp\left(\psi \right)\right)}^{a}}{{\left(1+\exp\left(\psi \right)\right)}^{b}}=2^{-b}\exp\left(\kappa \psi \right)\int_{0}^{\infty }\exp\left(-\omega_{i}\psi^{2}/2\right)p\left(\omega_{i}\right)d\omega_{i} \end{array}$$
where κ=a−b/2. The above identity implies that the posterior distribution of the coefficients can be represented as a multivariate normal distribution:(22)$$\begin{array}{c} \theta_{\gamma }∼N\left(\Lambda_{\gamma }^{-1}\xi ,\Lambda_{\gamma }^{-1}\right) \end{array}$$
where $\xi =J_{\gamma }^{T}\kappa$, $\Lambda_{\gamma }=J_{\gamma }^{T}WJ_{\gamma }+V_{\gamma }^{-1}$, κ is an *n*-dimensional vector with entries $\kappa_{i}=y_{i}-1/2$ and *W* is a diagonal matrix with ω appearing along its diagonal. By integrating out coefficient $\theta_{\gamma }$, analytically conditioned on Pólya-gamma random variables ω, we obtain the conditional marginal likelihood
(23)$$\begin{array}{c} p\left(y|\gamma ,\omega \right)\propto |V_{\gamma }{|}^{-\frac{1}{2}}{\left|\Lambda_{\gamma }\right|}^{-\frac{1}{2}}\exp\left\{\frac{1}{2}\xi^{T}\Lambda_{\gamma }^{-1}\xi \right\}. \end{array}$$

In each iteration of the MCMC algorithm, we update γ and ω alternatively. To refresh ω, we can perform a simulation directly from its posterior distribution, which also follows a Pólya-gamma distribution given by
(24)$$\begin{array}{c} \omega_{i}∼\mathrm{PG}\left(1,\eta_{\gamma ,i}\right). \end{array}$$
where linear predictor $\eta_{\gamma ,i}$ involves coefficient $\theta_{\gamma }$ simulated from (Equation 22). Efficient samples from the Pólya-gamma random variables can be simulated using the R package pgdraw (version 1.1) [56]. In addition, Zens et al. described the ultimate Pólya-gamma sampler [57] to address the slow mixing rate for categorical imbalanced data, as illustrated in [58].

In general, the data-augmentation schemes may not be applicable to all generalised linear models and survival models. Specifically, for the Cox proportional hazards with partial likelihood or the Weibull model, there is currently no suitable data augmentation to directly yield a parametric posterior distribution for the regression coefficients.

### 3.2. Laplace Approximation

Assuming a unimodal posterior distribution of the regression coefficients, the Laplace approximation estimates the marginal likelihood with a second-order Taylor approximation. This method leads to a Gaussian integral, with the solution of the marginal likelihood being given by
(25)$$\begin{array}{c} p_{\mathrm{LA}}\left(y|\gamma \right)=p\left(y|{\hat{\theta }}_{\gamma },\gamma \right)p\left({\hat{\theta }}_{\gamma }|\gamma \right){\left|{\hat{H}}_{\gamma }\right|}^{-\frac{1}{2}}{\left(2\pi \right)}^{\frac{p_{\theta_{\gamma }}}{2}} \end{array}$$
where ${\hat{\theta }}_{\gamma }$ is the posterior mode of $\theta_{\gamma }$ and $H_{\gamma }$ is the negated Hessian of $\log p\left(y|\theta_{\gamma },\gamma \right)+\log p\left(\theta_{\gamma }|\gamma \right)$ evaluated at mode ${\hat{\theta }}_{\gamma }$. Additionally, the Laplace approximation provides a normal approximation to the posterior distribution of coefficient $\theta_{\gamma }$ as
(26)$$\begin{array}{c} \pi_{\mathrm{LA}}\left(\theta_{\gamma }\right)=N_{p_{\theta_{\gamma }}}\left({\hat{\theta }}_{\gamma },{\hat{H}}_{\gamma }^{-1}\right). \end{array}$$
To incorporate the Laplace approximation in MH sampling, we replace marginal likelihood p(y|γ) in (Equation 19) with the approximate $p_{\mathrm{LA}}\left(y|\gamma \right)$ as described above.

Laplace approximation has been shown to be asymptotically consistent for estimating Bayes factors [59] and Bayesian variable selection on generalised linear models [60]. In finite-sample problems, however, Laplace approximation introduces biases, so $p_{\mathrm{LA}}\left(y|\gamma \right)$ is not an unbiased estimate of true marginal likelihood p(y|γ). The resulting MCMC scheme, which involves the step of Laplace approximation, targets a different distribution compared with the true posterior π(γ). Instead, it targets the distribution $\pi_{\mathrm{LA}}\left(\gamma \right)\propto p_{\mathrm{LA}}\left(y|\gamma \right)p\left(\gamma \right)$.

### 3.3. Correlated Pseudo-Marginal Method

We can alternatively make use of normal approximation $\pi_{\mathrm{LA}}\left(\theta_{\gamma }\right)$ to derive an importance sampling estimate of marginal likelihood p(y|γ). This estimator is unbiased and given by
(27)$$\begin{array}{c} \hat{p}\left(\gamma |y\right)=\frac{1}{N}\sum_{i=1}^{N}\frac{p\left(y|\theta_{\gamma }^{\left(i\right)},\gamma \right)p\left(\theta_{\gamma }^{\left(i\right)}|\gamma \right)}{\pi_{\mathrm{LA}}\left(\theta_{\gamma }^{\left(i\right)}\right)} \end{array}$$
where $\theta_{\gamma }^{\left(1\right)},\cdots ,\theta_{\gamma }^{\left(N\right)}$ are *N* samples from $\pi_{\mathrm{LA}}\left(\theta_{\gamma }\right)$. As in Laplace approximation, we can replace marginal likelihood p(y|γ) in (Equation 19) with estimated marginal likelihood $\hat{p}\left(y|\gamma \right)$. This leads to the pseudo-marginal scheme in [61,62]. Andrieu and Roberts [62] show that the resulting Markov Chain preserves π-reversibility as long as estimated marginal likelihood $\hat{p}\left(y|\gamma \right)$ is an unbiased estimator of the true marginal likelihood, p(y|γ).

It is possible to extend a pseudo-marginal method to a correlated pseudo-marginal method [21], with the aim of reducing the estimation variance of the ratio of estimated marginal likelihoods $\hat{p}\left(y|\gamma^{\prime }\right)/\hat{p}\left(y|\gamma \right)$. The correlated pseudo-marginal method is applied to Bayesian variable selection for the logistic regression model in [27], which provides an implementation that we also adopt in this work.

### 3.4. Approximate Laplace Approximation

The above Laplace approximation and correlated pseudo-marginal methods are computationally intensive due to the optimisation process required to obtain the normal approximation in (Equation 26), especially for dealing with large-*n* data. To avoid the overwhelming computational cost associated with the optimisation process, Rossell et al. [16] introduce the approximate Laplace approximation method (ALA), which is more computationally tractable for large-*n* problems. In this work, we consider the alternative formula described in supplementary material S.1. of [16], as it offers better computational stability when inverting the Hessian under the independent prior in (Equation 7).

In ALA, a Taylor expansion of log-posterior density $\log p\left(y|\theta_{\gamma },\gamma \right)+\log p\left(\theta_{\gamma }|\gamma \right)$ is performed at initial value $\theta_{\gamma }^{0}$. Solving the resulting Gaussian integral leads to
(28)$$\begin{array}{c} p_{\mathrm{ALA}}\left(y|\gamma \right)=p\left(y|\theta_{\gamma }^{0},\gamma \right)p\left(\theta_{\gamma }^{0}|\gamma \right){\left(2\pi \right)}^{\frac{d}{2}}{\left|H_{\gamma }^{0}\right|}^{-\frac{1}{2}}\exp\left\{\frac{1}{2}g_{\gamma }^{0T}{\left(H_{\gamma }^{0}\right)}^{-1}g_{\gamma }^{0}\right\} \end{array}$$
where $g_{\gamma }^{0}$ and $H_{\gamma }^{0}$ are the gradient and Hessian of the negative log-posterior density evaluated at $\theta_{\gamma }^{0}$, respectively. It is suggested in [16] to set initial value $\theta_{\gamma }^{0}$ to $\theta_{\gamma }^{0}=0$ for convenience.

By applying ALA to the MH acceptance probability in (Equation 19), we obtain an MCMC algorithm that targets the ALA posterior distribution $\pi_{\mathrm{ALA}}\left(\gamma \right)\propto p_{\mathrm{ALA}}\left(y|\gamma \right)p\left(\gamma \right)$ as the equilibrium distribution. Although Ref. [16] shows that ALA can recover the optimal model with respect to a mean squared loss, it is important to note that ALA is not consistent with respect to the marginal likelihood (in contrast to the classical Laplace approximation) and $p_{\mathrm{ALA}}\left(y|\gamma \right)$ is not an unbiased estimator of the true marginal likelihood, p(y|γ).

## 4. Point-Wise Implementation of Adaptive Random Neighbourhood Informed Proposal

### 4.1. The PARNI Proposal

The PARNI proposal belongs to the class of random neighbourhood informed proposals, which typically involve the following two steps: (i) sampling a neighbourhood N∼p(·|γ) and then (ii) proposing a model $\gamma^{\prime }$ within this neighbourhood N according to the informed proposal of [31]. In the PARNI proposal, we assume that the randomness in neighbourhood generation is characterised by an auxiliary variable k∈K, with conditional distribution p(k|γ), which leads to neighbourhood N=N(γ,k), such that p(k|γ)=p(N|γ). By defining $K={\left\{0,1\right\}}^{p}$, the value of *k* indicates whether the change in the corresponding position in γ is included in neighbourhood N. Specifically, for those positions *j* such that $k_{j}=1$, the neighbourhood consists of models obtained by varying some or all of these positions in the current model, γ.

The conditional distribution of *k* takes the product form $p\left(k|\gamma \right)=\prod_{j=1}^{p}p\left(k_{j}|\gamma_{j}\right)$, where each $k_{j}$ depends on the corresponding component $\gamma_{j}$ in γ. This probability distribution is driven by a set of tuning parameters $\left(A_{1},\cdots ,A_{p},D_{1},\cdots ,D_{p}\right)$, where $A_{j},D_{j}\in \left(\epsilon ,1-\epsilon \right)$ for a small value of ϵ∈(0,1/2). The probabilities of event $k_{j}=1$ are then defined by
(29)$$\begin{array}{c} p\left(k_{j}=1|\gamma_{j}=0\right)=A_{j},\;p\left(k_{j}=1|\gamma_{j}=1\right)=D_{j}, \end{array}$$
and the consequent neighbourhood is constructed as
(30)$$\begin{array}{c} N\left(\gamma ,k\right)=\left\{\gamma^{*}\in \Gamma |\;\gamma_{j}^{*}=\gamma_{j}\;\;\forall j\;\;s.t.\;\;k_{j}=0\right\}. \end{array}$$
Neighbourhood N(γ,k) contains $2^{p_{k}}$, models where $p_{k}$ denotes the number of 1s in *k*. Performing a full enumeration over the entire neighbourhood is, therefore, computationally expensive when $p_{k}$ is large. In fact, it becomes computationally infeasible to explore the whole neighbourhood when $p_{k}$ is beyond 30. Liang et al. [33], therefore, consider a point-wise approximate implementation of this algorithm that dramatically reduces the number of model probability evaluations from $O(2^{p_{k}})$ to $O(2p_{k})$.

The point-wise implementation proceeds by constructing a sequence of smaller neighbourhoods $\left\{N_{r}\right\}$ such that each $N_{r}$ is a subset of N. A proposed model $\gamma^{\prime }$ is sequentially simulated from these neighbourhoods $\left\{N_{r}\right\}$ according to locally informed proposals $q_{N_{r}}$. This procedure requires us to define a sequence of intermediate models $\gamma =\gamma \left(0\right)\to \gamma \left(1\right)\to \cdots \to \gamma \left(p_{k}\right)=\gamma^{\prime }$. We collect positions *j* such that $k_{j}=1$ and define them as $j_{1},\cdots ,j_{p_{k}}$ (the order is random). Small neighbourhood $N_{r}$ is then defined as follows:(31)$$\begin{array}{c} N_{r}=N\left(\gamma \left(r-1\right),j_{r}\right)=\left\{\gamma^{*}\in \Gamma |\;\gamma_{j}^{*}=\gamma {\left(r-1\right)}_{j}\;\;\forall j\neq j_{r}\right\}. \end{array}$$
Each small neighbourhood $N_{r}$ only consists of two models, γ(r−1) and $\gamma^{*}$, which only differ with γ(r−1) at position $j_{r}$. The resulting proposal mass function is
(32)$$\begin{array}{c} q_{k}\left(\gamma ,\gamma^{\prime }\right)=\prod_{r=1}^{p_{k}}q_{N_{r}}\left(\gamma \left(r-1\right),\gamma \left(r\right)\right) \end{array}$$
where $q_{N}$ is the locally informed proposal over neighbourhood N and is defined by
(33)$$\begin{array}{c} q_{N}\left(\gamma ,\gamma^{\prime }\right)\propto \left\{\begin{array}{cc} g\left(\frac{\pi \left(\gamma^{\prime }\right)p\left(k|\gamma^{\prime }\right)}{\pi (\gamma )p(k|\gamma )}\right){\left(\frac{\zeta }{1-\zeta }\right)}^{d_{H}\left(\gamma ,\gamma^{\prime }\right)}, & \mathrm{if}\;\gamma^{\prime }\in N\\ 0, & \mathrm{otherwise}. \end{array}\right. \end{array}$$
Tuning parameter ζ∈(ϵ,1−ϵ) denotes the non-informative jumping probability. Two different methods for adapting ζ are provided in [33]. One of the key factors influencing the performance of the informed proposal in (Equation 33) is the choice of weighting function *g*. Given a positive real number x>0, a balancing function is defined as a function *g* that satisfies the condition g(x)=xg(1/x). The locally informed proposal constructed with a balancing function is locally optimal in terms of Peskun ordering under mild conditions [31]. For the comparisons between different balancing functions, see Supplement B.1.3 of [31]. In this work, we exclusively focus on the Hastings’ choice of balancing function given by $g_{H}\left(x\right)=\min\left\{1,x\right\}$, as $g_{H}$ has demonstrated better empirical performance in many problems (e.g., [33]).

To construct a π-reversible chain in the MH scheme, we define a collection of neighbourhoods $\left\{N_{r}^{\prime }\right\}$ for the reverse moves, where $\left\{N_{r}^{\prime }\right\}$ are identical to $\left\{N_{r}\right\}$ but with reverse order. For a more detailed explanation of the PARNI proposal, we refer to Section 4.2.1 of [33]. The MH acceptance probability of the PARNI proposal is given by
(34)$$\begin{array}{cc} \alpha (\gamma ,\gamma^{\prime }) & =\min\left\{1,\frac{\pi \left(\gamma^{\prime }\right)p\left(k|\gamma^{\prime }\right)q_{k}\left(\gamma^{\prime },\gamma \right)}{\pi \left(\gamma \right)p\left(k|\gamma \right)q_{k}\left(\gamma ,\gamma^{\prime }\right)}\right\}\\ \; & =\min\left\{1,\frac{\hat{p}\left(y|\gamma^{\prime }\right)p\left(\gamma^{\prime }\right)p\left(k|\gamma^{\prime }\right)q_{k}\left(\gamma^{\prime },\gamma \right)}{\hat{p}\left(y|\gamma \right)p\left(\gamma \right)p\left(k|\gamma \right)q_{k}\left(\gamma ,\gamma^{\prime }\right)}\right\}. \end{array}$$

**Remark 1.** 
*The concept of a neighbourhood is also used in other schemes designed to estimate discrete posterior distributions, including the Shotgun Stochastic Search (SSS) approach [34] and Hamming ball sampler (HBS) [28]. The SSS method works on the same neighbourhood as that constructed with the add–delete–swap proposal [22]. Given the current model, γ, SSS constructs a neighbourhood N(γ) that comprises three disjoint sub-neighbourhoods: $N_{a}\left(\gamma \right)$, $N_{d}\left(\gamma \right)$ and $N_{s}\left(\gamma \right)$. The “addition” neighbourhood, $N_{a}\left(\gamma \right)$, is formed by adding a covariate into the model, and similarly, the “deletion” neighbourhood, $N_{d}\left(\gamma \right)$, is formed by deleting a covariate from the model. Lastly, $N_{s}\left(\gamma \right)$ is obtained by swapping an included covariate with an excluded one. On the contrary, the HBS constructs neighbourhoods based on the Hamming ball, $H_{d}\left(\gamma \right)$, consisting of models that differ from γ by at most d positions. The typical example is the 1-Hamming ball, denoted by $H_{1}\left(\gamma \right)$. It is worth mentioning that the SSS and HBS approaches construct neighbourhoods with sizes of $(p_{\gamma }+1)p$ and p, respectively. By contrast, the PARNI proposal constructs neighbourhoods that are typically approximately of size $p_{\gamma^{*}}$, where $\gamma^{*}$ denotes the true underlying model. Assuming that the size of the true underlying model is much smaller than p, as is typical in many applications, $p_{\gamma^{*}}\ll p$, and PARNI exhibits a higher level of scalability in handling the large-p data in comparison to SSS and HBS.*


In the remaining parts of this section, we will describe a novel scheme to estimate tuning parameters *A* and *D*, and a new method for efficiently estimating the marginal likelihood in the locally informed proposal of (Equation 33).

### 4.2. New Adaptation Scheme on Algorithmic Tuning Parameters

The performance of the PARNI proposal is largely dictated by the choice of algorithmic tuning parameters *A* and *D*. Griffin et al. [26] consider the informed proposal of the form
(35)$$\begin{array}{c} A_{j}=\min\left\{1,\frac{\pi_{j}}{1-\pi_{j}}\right\},\;D_{j}=\min\left\{1,\frac{1-\pi_{j}}{\pi_{j}}\right\} \end{array}$$
where $\pi_{j}$ denotes the PIP for the *j*-th covariate and is defined by $\pi_{j}=\pi \left(\gamma_{j}=1\right)$. In their ASI scheme for BVS in the linear regression model, tuning parameters π are adaptively updated based on a Rao-Blackwellised estimate of the PIP given in Equation (Equation 10) of [26]. Wan and Griffin [27] extend ASI to the logistic regression model, in which they derive Rao-Blackwellised estimates of PIPs conditioned on the Pólya-gamma latent variables. Generalising this method to other generalised linear models and survival models that lack a suitable data-augmentation scheme is challenging. As the analytic marginal likelihood is inaccessible, it becomes intractable to derive Rao-Blackwellised estimates of PIPs. As an alternative, a simple Monte Carlo average over the output ${\left\{\gamma^{\left(l\right)}\right\}}_{l=1}^{L}$ can be taken, where *L* is the current iteration number. This ergodic average is calculated as
(36)$$\begin{array}{c} {\tilde{\pi }}_{j}^{\left(L\right)}=\frac{1}{L}\sum_{l=1}^{L}I\left\{\gamma_{j}^{\left(l\right)}=1\right\}. \end{array}$$
The ergodic average tends to be broad and biased, and it often downweights the importance of highly correlated covariates. Using the ergodic average directly in the PARNI proposal, however, results in a feedback effect, wherein a poor ergodic average leads to inadequate exploration over the sample space, leading to a subsequent bad ergodic average. To combat this phenomenon, we consider the following composition of two measures: a “warm-start” approximation, ${\tilde{\pi }}^{\left(0\right)}$, and the ergodic average, ${\tilde{\pi }}^{\left(L\right)}$, obtained from the first *L* samples. This composite estimate is adaptively updated using the formula
(37)$$\begin{array}{c} {\hat{\pi }}_{j}^{\left(L\right)}=\phi_{L}{\tilde{\pi }}_{j}^{\left(0\right)}+\left(1-\phi_{L}\right){\tilde{\pi }}_{j}^{\left(L\right)} \end{array}$$
where ${\left\{\phi_{l}\right\}}_{l=1}^{L}$ is a set of weights that control the trade-off between the warm-start approximation and the ergodic average.

Warm-start approximation ${\tilde{\pi }}_{j}^{\left(0\right)}$ is computed in the following way: Given the initial model of the Markov Chain, $\gamma^{\left(0\right)}$, and two related models, $\gamma^{j↑}=\left(\gamma_{j}=1,\gamma_{-j}^{\left(0\right)}\right)$ and $\gamma^{j↓}=\left(\gamma_{j}=0,\gamma_{-j}^{\left(0\right)}\right)$, for the *j*-th component, the Rao-Blackwellised estimate of the *j*-th PIP at model $\gamma^{\left(0\right)}$ is given by
$$\begin{array}{cc} P(\gamma_{j}=1|\gamma_{-j}=\gamma_{-j}^{\left(0\right)},y) & =\frac{\pi (\gamma^{j↑})}{\pi \left(\gamma^{j↑}\right)+\pi \left(\gamma^{j↓}\right)}=\frac{\frac{p\left(y|\gamma^{j↑}\right)p\left(\gamma^{j↑}\right)}{p\left(y|\gamma^{j↓}\right)p\left(\gamma^{j↓}\right)}}{1+\frac{p\left(y|\gamma^{j↑}\right)p\left(\gamma^{j↑}\right)}{p\left(y|\gamma^{j↓}\right)p\left(\gamma^{j↓}\right)}}. \end{array}$$We consider the ALA in (Equation 28) initialised at the origin to estimate the intractable Bayes factor, $p\left(y|\gamma^{j↑}\right)/p\left(y|\gamma^{j↓}\right)$, for including the *j*-th covariate. Let $\eta_{i}$ be the *i*-th linear predictor, $\eta_{i}^{0}$ be the *i*-th linear predictor evaluated at the origin (i.e., $\eta_{i}^{0}=0$), $X_{j}$ denote the *j*-th column of data matrix *X*, $\tilde{y}$ be a vector with *i*-th component equal to $\partial p\left(y|\theta_{\gamma },\gamma \right)/\partial \eta_{i}$ evaluated at $\eta_{i}=\eta_{i}^{0}$ and *W* be a matrix such that $W_{il}=-\partial^{2}p\left(y|\theta_{\gamma },\gamma \right)/\partial \eta_{i}\partial \eta_{l}$ evaluated at $\eta_{i}=\eta_{i}^{0}$ and $\eta_{l}=\eta_{l}^{0}$. Thanks to the Schur complement, we can facilitate the computation of *p* Bayes factors as in [26,27]: when $\gamma^{\left(0\right)}=\gamma^{j↓}$,
(38)$$\begin{array}{c} \frac{\tilde{p}\left(y|\gamma^{j↑}\right)}{\tilde{p}\left(y|\gamma^{j↓}\right)}=d_{j}^{↑-\frac{1}{2}}g^{-\frac{1}{2}}\exp\left\{\frac{1}{2d_{j}^{↑}}\left({\tilde{y}}^{T}X_{\gamma }\Lambda_{\gamma }^{-1}X_{\gamma }^{T}WX_{j}-{\tilde{y}}^{T}X_{j}\right)\right\} \end{array}$$
where $\Lambda_{\gamma }=X_{\gamma }WX_{\gamma }+V_{\gamma }^{-1}$ and $d_{j}^{↑}=X_{j}^{T}WX_{j}+1/g-X_{j}^{T}WX_{\gamma }\Lambda_{\gamma }^{-1}X_{\gamma }^{T}WX_{j}$; when $\gamma^{\left(0\right)}=\gamma^{j↑}$
(39)$$\begin{array}{c} \frac{\tilde{p}\left(y|\gamma^{j↑}\right)}{\tilde{p}\left(y|\gamma^{j↓}\right)}=d_{j}^{↓-\frac{1}{2}}g^{-\frac{1}{2}}\exp\left\{-\frac{1}{2}d_{j}^{↓}{\left({\tilde{y}}^{T}X_{\gamma }{\left(\Lambda_{\gamma }^{-1}\right)}_{\cdot ,q+p_{j}}\right)}^{2}\right\} \end{array}$$
where $d_{j}^{↓}=1/{\left(\Lambda_{\gamma }^{-1}\right)}_{q+p_{j},q+p_{j}}$ and $p_{j}$ is the ordered position of the *j*-th variable. When working with data that involve a high level of collinearity, it is also possible to increase the number of the ALA Rao-Blackwellised estimates.

The last building block of adapting ${\hat{\pi }}_{j}^{\left(L\right)}$ is defining weight $\phi_{l}$. We employ a straightforward construction of $\phi_{l}$ given by
(40)$$\begin{array}{c} \phi_{l}=\left\{\begin{array}{cc} 1-\frac{1}{2}{\left(N_{b}-l+1\right)}^{-0.5} & \mathrm{if}\;l\leq N_{b}\\ \frac{1}{2}{\left(l-N_{b}\right)}^{-0.5} & \mathrm{if}\;l>N_{b}. \end{array}\right. \end{array}$$
where $N_{b}$ denotes the length of the burn-in period. This choice results in a weight that exceeds 1/2 during the period of burn-in and drops below 1/2 afterwards. Consequently, the PARNI proposal initially relies on the warm-start approximation to explore the model space. As the chain converges to the high-probability region and the ergodic average stabilises, the PARNI proposal gradually uses more information from the ergodic average. After running for a longer time, the PARNI proposal completely relies on the ergodic average.

### 4.3. The Adaptive ALA Informed Proposal

In each iteration of the PARNI proposal, the locally informed proposal in (Equation 33) relies on computing the posterior model probabilities. Using the estimates from the Laplace approximation or correlated pseudo-marginal scheme in PARNI can be computationally intractable in “large-*p*, large-*n*” situations due to the use of an optimisation algorithm that most run many times in one iteration. It should be noted, however, that the model probabilities in the locally informed proposal do not need to precisely match the true posterior model probabilities, and the PARNI proposal can still generate samples that preserve π-reversibility as long as the correct (or proper estimate) π is used in the MH acceptance probability of (Equation 34). In the locally informed proposal, one can incorporate the approximate Laplace approximation initialised at the origin to design the proposal distribution. In the MH acceptance probability, we can then use the estimates obtained from the Laplace approximation or correlated pseudo-marginal method. Based on empirical observations, however, this ALA informed proposal may not always mix well. One reason for this is the phenomenon of downweighting the model probabilities of non-null models in favour of the null model, resulting in an informed proposal that is less informative than the true likelihood. The simulated chain is, therefore, more likely to get stuck and becomes less effective in exploring the model space.

Alternatively, we can note that the ALA estimate coincides with the Laplace approximation when the initial value is chosen to be posterior mode ${\hat{\theta }}_{\gamma }$ under model $M_{\gamma }$. Therefore, the accuracy of the ALA estimate is crucially influenced by the choice of initial value $\theta_{\gamma }^{0}$. We employ the ALA informed proposal with an adaptive initial value for ALA (adaptive ALA), which aims to reduce the estimation errors and thus improve the overall performance of the MCMC algorithm.

For each model in the neighbourhood, the adaptive ALA starts with an initial guess of linear predictor η and proceeds with the following steps:Calculate a “guess” estimate from linear predictor η:
(41)$$\begin{array}{c} \theta_{\gamma }^{0}={\left(J_{\gamma }^{T}J_{\gamma }\right)}^{-1}J_{\gamma }^{T}\eta . \end{array}$$Perform one step of Newton’s method and obtain an updated estimate of the coefficient:
(42)$$\begin{array}{c} {\tilde{\theta }}_{\gamma }=\theta_{\gamma }^{0}-{\left(H_{\gamma }^{0}\right)}^{-1}g_{\gamma }^{0} \end{array}$$
where $g_{\gamma }^{0}$ and $H_{\gamma }^{0}$ are the gradient and Hessian of the negated log-posterior density evaluated at $\theta_{\gamma }^{0}$, respectively.Use the ALA estimate in (Equation 28) with ${\tilde{\theta }}_{\gamma }$ as the initial value to estimate marginal likelihood p(y|γ).

Practically speaking, we can skip step 1 with the matrix inverse operation in (Equation 41) and obtain ${\tilde{\theta }}_{\gamma }$ directly from the initial guess of linear predictor η. This simplification is followed by [63] and is given in Appendix C. This approach leads to a coherent computational scheme that is easy to implement. We adaptively update the initial-guess η according to
(43)$$\begin{array}{c} {\hat{\eta }}_{i}^{\left(L\right)}=\frac{1}{L}\sum_{l=1}^{L}{\hat{\eta }}_{\gamma^{\left(l\right)},i} \end{array}$$
where ${\hat{\eta }}_{\gamma ,i}=J_{\gamma }{\hat{\theta }}_{\gamma }$ is the “optimal” *i*-th linear predictor obtained from MAP estimate ${\hat{\theta }}_{\gamma }$ under model $M_{\gamma }$. By storing the MAP estimate obtained from the Laplace approximation or correlated pseudo-marginal scheme, we can compute the linear predictor without introducing additional computational costs.

In addition, we experimented with adapting coefficients $\hat{\beta }$ from the posterior samples and using the coefficients of the covariates selected by γ to navigate the ALA. This approach did not work well, however, because the posterior distribution of β differs significantly from the posterior distribution of β conditioned on model γ. In contrast, the linear predictors offer more stability, in the sense that they do not vary as much across different models.

Combining all of the above components, we have the PARNI proposal. The complete algorithm is outlined in Algorithm 1.
**Algorithm 1** The algorithmic pseudo-code of the Point-wise Adaptive Random Neighbourhood Sampler with Informed proposal (PARNI)Initialise the chain at $\gamma^{\left(0\right)}$ and compute ${\left\{{\tilde{\pi }}_{j}^{\left(0\right)}\right\}}_{j=1}^{p}$;**for** i=1 to i=N **do**  Sample $k∼p(\;\cdot \;|\gamma^{\left(i-1\right)})$ as in (Equation 29);  Set $\gamma \left(0\right)=\gamma^{\left(i-1\right)}$, $p_{k}=\sum_{j=1}^{p}k_{j}$ and define $j_{1},\cdots ,j_{p_{k}}$;  **for** r=1 to $r=p_{k}$ **do**    Construct $N_{r}$ as in (Equation 31) and estimate $p(y|\gamma^{*})$ for all $\gamma^{*}\in N_{r}$ as in Section 4.3;    Sample $\gamma \left(r\right)∼q_{N_{r}}\left(\gamma \left(r-1\right),\;\cdot \;\right)$ as in (Equation 33);  **end for**  Set $\gamma^{\prime }=\gamma \left(p_{k}\right)$, estimate $p(y|\gamma^{\prime })$ by LA or CPM and sample U∼Unif(0,1);  If $U<\alpha (\gamma^{\left(i-1\right)},\gamma^{\prime })$ as in (Equation 34), then $\gamma^{\left(i\right)}=\gamma^{\prime }$, else $\gamma^{\left(i\right)}=\gamma^{\left(i-1\right)}$;  **for** j=1 to j=p **do**    Update ${\tilde{\pi }}_{j}^{\left(i\right)}$ as in (Equation 36) and ${\hat{\pi }}_{j}^{\left(i\right)}$ as in (Equation 37);    Update $A_{j}^{\left(i\right)}=\min\left\{1,{\hat{\pi }}_{j}^{\left(i\right)}/\left(1-{\hat{\pi }}_{j}^{\left(i\right)}\right)\right\}$;    Update $D_{j}^{\left(i\right)}=\min\left\{1,\left(1-{\hat{\pi }}_{j}^{\left(i\right)}\right)/{\hat{\pi }}_{j}^{\left(i\right)}\right\}$;  **end for**  Update $\omega^{\left(i\right)}$ using the selected adaption scheme and ${\hat{\eta }}^{\left(i\right)}$ as in (Equation 43);**end for**

## 5. Experiments

### 5.1. Simulated Data-Sets with Adaptive ALA Informed Proposal

In this subsection, we study the mixing behaviour of different versions of the PARNI proposal for the logistic regression model, Cox PHs and Weibull survival models. We simulate two data-sets with 500 covariates and 500 observations as described in Appendix D and compare the following four algorithms:**PARNI-adaptiveALA:** The PARNI proposal with adaptive approximate Laplace approximation in the informed proposal and the correlated pseudo-marginal scheme in the MH acceptance probability.**PARNI-LA:** The PARNI proposal with Laplace approximation in the informed proposal and the correlated pseudo-marginal scheme in the MH acceptance probability.**PARNI-ALA:** The PARNI proposal with approximate Laplace approximation in the informed proposal and the correlated pseudo-marginal scheme in the MH acceptance probability.**ADS (thinned):** The PARNI proposal with approximate Laplace approximation in the informed proposal and the correlated pseudo-marginal scheme in the MH acceptance probability.

The first three algorithms were run for 10,000 iterations, with the first 2000 iterations being discarded as burn-in, whereas the ADS (thinned) proposal was run for a CPU time similar to that of PARNI-adaptiveALA and PARNI-ALA, with all collected samples being thinned to 10,000 values.

Figure 1 presents trace plots of the log-posterior model probability and bar plots of CPU time for the PARNI-adaptiveALA, PARNI-LA, PARNI-ALA and ADS (thinned) proposals in the logistic model, and the Cox PHs and Weibull models. In all three models, the PARNI-adaptiveALA proposal mixes as well as the PARNI-LA proposal and performs much better than the PARNI-ALA proposal. The result of the ADS (thinned) proposal provides the benchmark performance of a simple add–delete–swap MCMC scheme on these data-sets for comparison purposes. As illustrated in Figure 1, the adaptive ALA informed proposal is computationally much cheaper than the LA informed proposal. In comparison to the ALA informed proposal, the adaptive ALA informed proposal is also computationally competitive, and it only introduces the additional computational costs of updating linear predictor ${\hat{\eta }}^{\left(L\right)}$ and computing initial value ${\tilde{\theta }}_{\gamma }$ from the estimate of linear predictor ${\hat{\eta }}^{\left(L\right)}$ as in (Equation 42). In addition, the PARNI-adaptiveALA proposal demonstrates improved mixing behaviour in comparison to the baseline add–delete–swap proposal in all three models with similar CPU time. Therefore, we conclude that the PARNI-adaptiveALA proposal is more computationally efficient than the informed proposals constructed using Laplace approximation or ALA initialised at the origin.

### 5.2. Logistic Regression: Genetic Mapping for Systemic Lupus Erythematosus

Genetic mapping is a process of locating a specific gene or genetic variant within a particular genomic region and has the objective to find the precise genetic elements responsible for a particular trait or disease phenotype. One common application is to study whether an individual has a particular disease. In this scenario, one can use a logistic regression model with the response variable based on the case/control design and explanatory variables consisting of single-nucleotide polymorphisms (SNPs).

We consider a problem of identifying the SNPs that play a crucial role in predicting Systemic Lupus Erythematosus using a case/control study. It consists of genotypes from a genome-wide genetic case/control association study involving 4035 cases and 6959 controls, where the cases are SLE patients and the controls are from a public repository of European ancestry. These data were previously studied in [64] using step-wise logistic regression in a meta-analysis. In Chapter 5 of [65], Griffin and Steel apply Bayesian variable selection to analyse these SLE data but only focus on exploring the relationship between disease and SNPs on Chromosome 1. In addition to their work, we extend the study by including a total of four chromosomes. We consider a different number of SNPs for each chromosome, with Chromosome 1 having 5771 SNPs, Chromosome 3 having 42,430 SNPs, Chromosome 11 having 32,290 SNPs and Chromosome 21 having 9306 SNPs. We consider the prior specification in Section 2.2 with hyper-parameter g=1/4, $\sigma_{\alpha }^{2}=1$ and assume the hyper-prior of h∼Beta(1,(p−5)/5), where *p* denotes the number of SNPs. The full details of the data-set are provided in Table 1, including the five fixed covariates (gender and top four principal components of expressed genes) that are mandatory in all models.

We implement the following four algorithms:**PARNI-DA**: PARNI proposal with Pólya-gamma data augmentation in both the informed proposal and the MH acceptance probability.**PARNI-CPM**: PARNI proposal with adaptive ALA informed proposal and correlated pseudo-marginal method in the MH acceptance probability.**ADS-DA**: Add–delete–swap proposal with Pólya-gamma data augmentation in the MH acceptance probability.**ADS-CPM**: Add–delete–swap proposal with the correlated pseudo-marginal method in the MH acceptance probability.

These MCMC algorithms all simulate samples from the exact posterior distribution, π. We treat the ADS-DA proposal as the baseline to showcase the rapid mixing of the PARNI proposals. Each algorithm was run for 1 h with 10 repetitions, and we recorded the estimates of PIPs. Firstly, we calculated the mean squared errors of the estimates of *p* PIPs compared with the “gold standard” estimates taken from the PARNI-CPM proposal, which was run for roughly 12 h. Then, we took the average over *p* mean squared errors to obtain the average mean squared error (average MSE). To compare the computational efficiency of the PARNI proposals with the baseline ADS-DA proposal, we provide the relative efficiency (in brackets) as the ratio of their average MSE.

The average MSEs and relative efficiency values are presented in Table 2. The PARNI proposals consistently outperform the ADS proposals in terms of the average MSE. The PARNI proposals show at least twofold improvements over the add–delete–swap proposal and lead to much larger improvements in most cases, such as in Chromosome 21, where the PARNI proposals perform 78 times better than the add–delete–swap proposal. On the other hand, both the PARNI-DA and ADS-DA proposals consistently result in smaller average MSEs compared with the PARNI-CPM and ADS-CPM proposals due to their computational advantages. Firstly, data augmentation can evaluate the conditional marginal likelihood without finding the posterior mode using iteratively re-weighted least squares. Secondly, the Pólya-gamma latent variables are drawn using the R package pgdraw (version 1.1) [56] implemented using Rcpp (version 1.0.10) [66].

### 5.3. Survival Analysis: Variable Selection for Five Large
Cancer-Related Gene Expression Data-Sets

We consider a total of four cancer-related real data-sets, where the first two data-sets are for breast cancer and the remaining data-sets are for lung cancer. NKI Breast Cancer Data (https://data.world/deviramanan2016/nki-breast-cancer-data, accessed on 4 September 2023) contain patient info, treatment, survival time and the 1554 most varying genes of 272 breast cancer patients. These data were analysed in [67,68] with the aim of reducing the mortality rates from this disease. The METABRIC breast cancer data-set is derived from the Molecular Taxonomy of Breast Cancer International Consortium (METABRIC) database. The METABRIC data-set was analysed in [69,70] and is publicly available in [71] (https://www.cbioportal.org/study/summary?id=brca_metabric, accessed on 4 September 2023). The data contain 1907 patients with the gene expression for 331 genes and mutations for 175 genes. Gene mutation variables are encoded as 1 if a mutation exists and 0 otherwise. For both data-sets, we include some clinical covariates, including the age of the patients and the stage of the cancer, as suggested by [72]. We also consider the treatment variables (such as chemotherapy and surgery type), which also influence survival time. The last two lung cancer data-sets, “GSE31210” and “GSE4573”, were previously studied in [72], and they are publicly available in the Gene Expression Omnibus repository [73]. See Figure 1 in [72] for the estimated survival functions of these three data-sets. We provide the full details of these four real data-sets in Table 3.

We consider two computational algorithms used in previous studies for logistic models, the PARNI-CPM and ADS-CPM proposals, as a data augmentation scheme is not available for the Cox PHs or Weibull model. We consider the hyper-prior of h∼Beta(1,(p−5)/5), where *p* denotes the number of genetic covariates, and impose a half-Cauchy hyper-prior on $\sqrt{g}$, where a Gibbs update is taken on *g* conditioned on the model (see Appendix A for more details). In addition, we assume $\sigma_{\alpha }^{2}=10^{5}$ and $\sigma_{k}^{2}=10^{5}$ (only for the Weibull model).

The average MSEs and relative efficiency values of the PARNI-CPM and ADS-CPM proposals on these four survival data-sets are shown in Table 4. For the Weibull model, the PARNI proposal consistently exhibits better computational efficiency compared with add–delete–swap on all four data-sets. In the case of the NKI and METABRIC data-sets, which have a relatively small number of covariates, the PARNI-CPM proposal produces PIP estimates that are seven times more accurate compared with ADS-CPM. For high-dimensional data-sets, we can obtain PIP estimates from the PARNI-CPM proposal that are two times better than ADS-CPM. The lesser improvement observed in the high-dimensional examples can be attributed to the increasing number of unimportant covariates, where both algorithms are good at excluding these unimportant covariates from the models.

The Bayesian variable selection in the Cox PHs model with partial likelihood is generally more challenging compared with the Weibull model. The primary reason is that the inclusion of the non-parametric setup introduces additional complexities in evaluating the log-likelihood functions and its Hessian matrices. The PARNI-CPM proposal provides roughly two times better estimates on the NKI and GSE4573 data-sets compared with ADS-CPM. However, the ADS-CPM proposal shows better performance on the remaining two data-sets. In the “small-*p*, large-*n*” METABRIC data, the add–delete–swap proposal shows greater computational efficiency compared with the PARNI proposal, as the informed proposal needs to evaluate many computationally expensive Hessian matrices. In fact, the computation of evaluating the Hessian matrix scales with the order of $O(n^{2})$, in contrast with parametric models, where the computation of the Hessian matrix scales linearly with *n*. In the GSE31210 data with few strong signals, the posterior distribution on model space is relatively flat, and both algorithms have smaller average MSEs compared with the other data-sets. In particular, the add–delete–swap proposal can run for more iterations; it is, therefore, more computationally efficient compared with the PARNI-CPM proposal.

## 6. Discussion

In this work, we apply the PARNI proposal to Bayesian variable selection problems in generalised linear models and survival models. We find that the informed proposal obtained from the approximate Laplace approximation with our new adaptive initial point yields improved efficiency and accuracy in posterior sampling. We compare the performance of the PARNI proposal with the baseline add–delete–swap proposal in numerous “large-*p*, large-*n*” real-world data-sets, and the PARNI proposal with the correlated pseudo-marginal method provides PIP estimates with smaller mean squared errors than the add–delete–swap proposal in most of the problems. The numerical results from the Cox PHs also provide useful insights to improve the PARNI proposal in the future. Code to run the PARNI proposal on the logistic regression model, and the Cox PHs and Weibull models is available at https://github.com/XitongLiang/The-PARNI-scheme.git (accessed on 4 September 2023).

In addition to the three models described in the paper, the proposed technique can be extended to other generalised linear models and survival models. Two possible extensions are the Gamma generalised linear model [74] and various Bayesian non-parametric approaches to survival analysis [75]. It is still a challenging problem to reduce the computational cost of simulating samples when a data-set contains a large number of observations. As highlighted in [76], simple sub-sampling strategies may not lead to a substantial improvement in the computational efficiency of posterior sampling. It would be interesting, therefore, to design an efficient PARNI scheme specifically tailored for large-*n* data-sets.

## Figures and Tables

**Figure 1 entropy-25-01310-f001:**
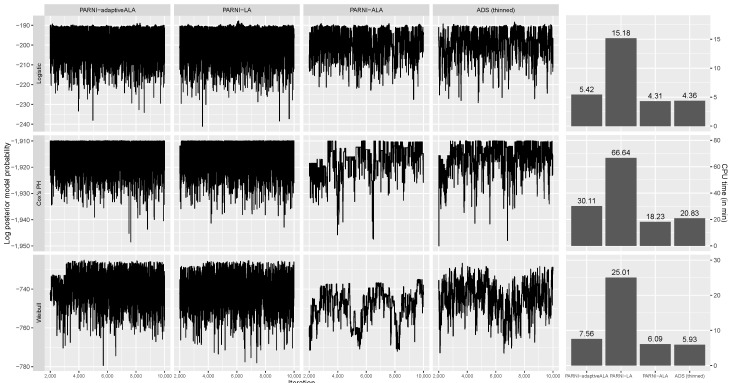
Left four columns: Trace plots of the log-posterior model probability from runs of the PARNI-adaptiveALA, PARNI-LA, PARNI-ALA and ADS (thinned) algorithms on simulated data-sets. Right column: Bar plots of the CPU time of simulating 10,000 samples on simulated data-sets with the PARNI-adaptiveALA, PARNI-LA, PARNI-ALA and ADS (thinned) algorithms.

**Table 1 entropy-25-01310-t001:** Details of Systemic Lupus Erythematosus data on Chromosomes 1, 3, 11 and 21.

Data-Set	Observations	Cases	Fixed Covariates	Genetic Covariates
Chromosome 1	10,995	4036	Gender, PC1–PC4	5771
Chromosome 3				42,430
Chromosome 11				32,290
Chromosome 21				9306

**Table 2 entropy-25-01310-t002:** Systemic Lupus Erythematosus data: The average mean squared errors of the ADS-DA, ADS-CPM, PARNI-DA and PARNI-CPM proposals in estimating the posterior inclusion probabilities of all SNPs (*smaller is better*). The relative efficiency as the ratio of the average MSE between algorithm A and the ADS-DA proposal presented in brackets (*larger is better*). The best performance is presented in **bold**.

Data-Set	Algorithms			
	ADS-DA	ADS-CPM	PARNI-DA	PARNI-CPM
Chromosome 1	$1.84\times 10^{-5}\;\left(1\right)$	$4.29\times 10^{-5}\;\left(0.43\right)$	$5.14\times 10^{-6}\;\left(3.58\right)$	$7.34\times 10^{-6}\;\left(2.51\right)$
Chromosome 3	$2.01\times 10^{-4}\;\left(1\right)$	$2.37\times 10^{-4}\;\left(0.85\right)$	$8.76\times 10^{-5}\;\left(2.30\right)$	$5.11\times 10^{-5}\;\left(3.94\right)$
Chromosome 11	$7.09\times 10^{-5}\;\left(1\right)$	$1.07\times 10^{-4}\;\left(0.66\right)$	$9.73\times 10^{-6}\;\left(7.29\right)$	$9.89\times 10^{-6}\;\left(7.15\right)$
Chromosome 21	$1.18\times 10^{-5}\;\left(1\right)$	$1.67\times 10^{-5}\;\left(0.71\right)$	$1.51\times 10^{-7}\;\left(78.08\right)$	$1.79\times 10^{-7}\;\left(65.93\right)$

**Table 3 entropy-25-01310-t003:** Details of 4 real data-sets for survival analysis.

Data-Set	Cancer Type	Observations	Events	Fixed Covariates	Genetic Covariates
NKI	Breast	272	77	Age, chemo, hormone, surgery, stage	1554
METABRIC	Breast	1903	622	Age, chemo, hormone, radio, surgery, stage	662
GSE31210	Lung	226	30	Age, gender, smoker, stage	54,675
GSE4573	Lung	130	63	Age, gender, stage	22,283

**Table 4 entropy-25-01310-t004:** Survival analysis data: The average mean squared errors of the ADS-CPM and PARNI-CPM proposals in estimating posterior inclusion probabilities of all genetic covariates (*smaller is better*). The relative efficiency as the ratio of the average MSE between algorithm A and the ADS-CPM proposal presented in brackets (*larger is better*). The best performance is presented in **bold**.

Data-Set	Cox PHs		Weibull Model	
	**ADS-CPM**	**PARNI-CPM**	**ADS-CPM**	**PARNI-CPM**
NKI	$5.54\times 10^{-4}\;\left(1\right)$	$3.19\times 10^{-4}\;\left(1.74\right)$	$3.21\times 10^{-5}\;\left(1\right)$	$4.33\times 10^{-6}\;\left(7.40\right)$
METABRIC	$1.27\times 10^{-3}\;\left(1\right)$	$3.80\times 10^{-3}\;\left(0.33\right)$	$1.36\times 10^{-3}\;\left(1\right)$	$1.73\times 10^{-4}\;\left(7.89\right)$
GSE31210	$1.26\times 10^{-6}\;\left(1\right)$	$3.56\times 10^{-6}\;\left(0.35\right)$	$3.91\times 10^{-5}\;\left(1\right)$	$2.16\times 10^{-5}\;\left(1.81\right)$
GSE4573	$4.57\times 10^{-5}\;\left(1\right)$	$2.54\times 10^{-5}\;\left(1.83\right)$	$3.56\times 10^{-5}\;\left(1\right)$	$8.37\times 10^{-6}\;\left(4.26\right)$

## Data Availability

Systemic Lupus Erythematosus was described in [64]. The NKI data-set is publicly available at https://data.world/deviramanan2016/nki-breast-cancer-data (accessed on 4 September 2023). The METABRIC data-set is publicly available at https://www.cbioportal.org/study/summary?id=brca_metabric (accessed on 4 September 2023). The GSE310210 AND GSE4573 are publicly available in the Gene Expression Omnibus repository [73]. The experiments were performed using R software, version 4.2.3 [77].

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
