# Peer review of "Adaptive MCMC for Bayesian Variable Selection in Generalised Linear Models and Survival Models"

_entropy, 2023, doi:10.3390/e25091310_

Round 1

Reviewer 2 Report

The authors presented an extended Point-wise implementation of Adaptive Random Neighborhood Informed proposal (PARNI) to efficiently sample models directly from the marginal posterior distribution Also, they describe an efficient and accurate estimation method for the marginal likelihood with adaptive parameters, and adapted of the PARNI by replacing the Rao-Blackwell estimates with the combination of a warm-start estimate and an ergodic average. The efficiency of the novel PARNI is demonstrated with several examples, and comparison to the baseline add-delete-swap proposal is given.

The task statement is clearly formulated. The authors well described and outlined the existing approaches  with a very detailed and high-quality review of the literature, and rigorously described suggested variable selection in GL models and survival analysis.

The approach clearly has a great advantage in tasks related to the study of the influence of genes on diseases. In addition, the proposed approach is provided with qualitative numerical examples.

Some phrases sound not well, ex “approximate Laplace approximation”.

I recommend to check text carefully.

But In general, the paper was done at a high level and may be published without additional review.

Some phrases sound not well, ex “approximate Laplace approximation”.

I recommend to check text carefully. But In general, the paper was done at a high level and may be published without additional review.

Reviewer 3 Report

See attached PDF.

There are no major problems or anything serious enough to cause confusion, but there are quite a few typos and grammatical mistakes throughout the manuscript. I haven't made a complete list, but there are at least 4 in the abstract alone.

Round 2

Reviewer 1 Report

Dear Authors,

Thanks for updating the manuscripts, and adding clarifications that improve the presentation of the paper and its readability. Please check that the link to the GitHub rep is working.

I do not have any other comment.

Reviewer 3 Report

The authors have made their code available on GitHub. It is undocumented, but the source files running the examples in the paper are straightforward enough to parse. My guess is that it could be used by R-literate users whose data is sufficiently similar to the examples considered here. English language has also been improved. 

The concern I raised about a potential lack of mixing is alleviated by the inclusion of trace plots.

I remain of the opinion that the paper is incremental. The software release is also unlikely to draw widespread interest in its current form; something akin to a documented R package would be much more likely to draw in practitioners. As it is, I think the work seems technically sound, but lacks a selling point to generate interest or warrant publication.